# Predictive Efficacy of the Perfusion Index for Hypotension following Spinal Anesthesia in Parturient Undergoing Elective Cesarean Section: A Systematic Review and Meta-Analysis

**DOI:** 10.3390/diagnostics13152584

**Published:** 2023-08-03

**Authors:** Kuo-Chuan Hung, Chien-Cheng Liu, Yen-Ta Huang, Jheng-Yan Wu, Jen-Yin Chen, Ching-Chung Ko, Chien-Ming Lin, Chung-Hsi Hsing, Ming Yew, I-Wen Chen

**Affiliations:** 1Department of Anesthesiology, Chi Mei Medical Center, Tainan City 71004, Taiwan; ed102605@gmail.com (K.-C.H.);; 2Department of Anesthesiology, E-Da Hospital, I-Shou University, Kaohsiung City 82456, Taiwan; 3Department of Nursing, College of Medicine, I-Shou University, Kaohsiung City 82445, Taiwan; 4School of Medicine, I-Shou University, Kaohsiung City 82445, Taiwan; 5Department of Surgery, National Cheng Kung University Hospital, College of Medicine, National Cheng Kung University, Tainan City 70101, Taiwan; 6Department of Nutrition, Chi Mei Medical Center, Tainan City 71004, Taiwan; 7Department of Medical Imaging, Chi Mei Medical Center, Tainan City 71004, Taiwan; 8Department of Health and Nutrition, Chia Nan University of Pharmacy and Science, Tainan City 71710, Taiwan; 9Department of Medical Research, Chi Mei Medical Center, Tainan City 71004, Taiwan; 10Department of Anesthesiology, Chi Mei Medical Center, Liouying, Tainan City 73657, Taiwan

**Keywords:** predictive efficacy, perfusion index, spinal anesthesia, hypotension, cesarean section

## Abstract

This meta-analysis assessed the predictive efficacy of perfusion index for hypotension following spinal anesthesia (SA) in parturients undergoing elective cesarean section (CS). Electronic databases, including Google Scholar, EMBASE, Cochrane Library, and MEDLINE, were searched from inception to June 2023. The primary outcome was the diagnostic accuracy of the perfusion index in predicting the probability of perioperative hypotension following SA. The review included 12 studies involving 2009 patients, published between 2017 and 2023. The pooled sensitivity and specificity were 0.81 (95% confidence interval (CI) = 0.72–0.87) and 0.75 (95% CI = 0.67–0.82), respectively. Additionally, the pooled area under the curve (AUC) was calculated as 0.84 (95% CI = 0.81–0.87), suggesting a moderate to good accuracy of the diagnostic test. Using Fagan’s nomogram plot, the positive likelihood ratio (LR) and negative LR were found to be 3 and 0.26, respectively. The results demonstrated that the perfusion index exhibited an acceptable level of accuracy in predicting perioperative hypotension after spinal anesthesia in parturients undergoing elective CS. These findings highlight the potential value of incorporating a perfusion index as a useful tool for clinicians to integrate into routine clinical practice, which necessitates further large-scale studies for verification.

## 1. Introduction

Spinal anesthesia is a commonly employed method for cesarean section owing to its advantages of reliability, rapid onset, and minimal transfer of drugs to the placenta [1,2]. However, spinal-anesthesia-induced sympathicolysis is frequently associated with a higher incidence of hypotension [3,4,5]. The incidence of hypotension varies depending on the clinical definition employed, with reported rates of up to 74.1% [6]. Post-spinal hypotension can have a negative impact on parturient and fetal outcomes [7]. For parturients, hypotension can lead to nausea and vomiting, which may be caused by reduced blood flow to the brainstem and vomiting center [8,9]. Furthermore, hypotension can result in inadequate placental perfusion, leading to fetal acidosis and lower Apgar scores [10]. Management focusing on preventing hypotension instead of reactively following its occurrence is recommended [1]. Although prophylactic vasopressors are commonly used to prevent the development of post-spinal hypotension [11,12], routine vasopressor use can also potentially lead to hypertension, which may be unfavorable in parturients with pre-existing cardiovascular diseases [10]. To improve patient outcomes and minimize the occurrence of hypotensive events, enhancing the accuracy of predicting post-spinal hypotension and implementing preventive measures are vital in clinical practice.

Although post-spinal hypotension is primarily caused by preganglionic sympathetic fiber blockade, preoperative peripheral vascular tone may also play a significant role in determining the susceptibility to hypotension. Several studies have reported that pregnant women, particularly after 30 weeks of gestation, tend to have more blood volume trapped in the extremities because of a decrease in pregnancy-associated vascular tone [13,14]. This physiological change further contributes to an increased risk of hypotension during spinal anesthesia in pregnant women [15]. Accordingly, baseline vascular tone monitoring may be an alternative method for early prediction of post-spinal hypotension risk. Photoplethysmography (PPG) is commonly used to monitor the oxygen levels in medical settings. The perfusion index, which is a derived parameter calculated from the PPG waveform, quantifies the relative amplitude of the pulsatile component of the PPG signal compared to the non-pulsatile or baseline component [16]. The perfusion index reflects the interaction between peripheral and central hemodynamic factors, including the baseline vascular tone and stroke volume [16]. Several studies have shown the potential of the perfusion index in predicting hypotension risk following spinal anesthesia [15,17,18]. Despite the potential benefits of the perfusion index in predicting post-spinal hypotension, systematic evaluation of its efficacy is lacking. Therefore, this meta-analysis aimed to assess the effectiveness of the perfusion index in predicting post-spinal hypotension in parturients undergoing cesarean section under spinal anesthesia.

## 2. Materials and Methods

### 2.1. Study Protocol

This review followed the PRISMA-DTA guidelines and was registered in PROSPERO (CRD42023433120).

### 2.2. Data Source and Literature Search

The retrieval of potentially eligible articles was performed by searching electronic databases, including MEDLINE, EMBASE, Cochrane Library, and Google Scholar, from their inception to 7 June 2023. The search was performed using the following keywords: (“Cesarean delivery” or “Cesarean section” or “Cesarean section” or “Abdominal deliveries” or “C-section” or “Postcesarean section”) and (“Postspinal” or “Spinal block” or “Subarachnoid block” or “Subarachnoid anesthesia” or “Intrathecal anesthesia” or “Intrathecal block” or “Neuraxial anesthesia” or “Spinal-epidural anesthesia” or “Intraspinal anesthesia”), and “Perfusion index” and (“Hypotension” or “Low blood pressure”). In addition to keyword searches, controlled vocabulary search terms, including EMBASE (Emtree) and Medline (MeSH), were employed to enhance the comprehensiveness of the literature search. To identify additional potentially eligible articles, a manual search of the reference lists of the obtained studies was performed. No restrictions were imposed on the country or language of publication. Table A1, Appendix A provides a detailed description of the search strategy used for MEDLINE.

### 2.3. Inclusion and Exclusion Criteria

The inclusion of studies in this meta-analysis was determined based on the following criteria. First, the studies included parturients who underwent elective cesarean section under spinal anesthesia, with or without a combination of epidural anesthesia. Second, studies have utilized the perfusion index to predict the risk of hypotension. Third, these studies provided detailed information on the sensitivity, specificity, and number of patients with hypotension. Various types of studies were considered eligible for this meta-analysis, including randomized controlled trials, cohort studies, and case–control studies.

Studies that met any of the following criteria were excluded from the analysis: first, studies solely reported as case series, abstracts, case reports, conference papers, or review articles were excluded. Second, studies that focused on parturients who underwent cesarean section under general or epidural anesthesia were excluded. Third, studies that lacked information related to hypotension or perfusion index were not considered. Finally, studies for which full-text versions were unavailable were excluded from the analysis.

### 2.4. Data Extraction

Two researchers conducted independent data extraction from individual studies and any discrepancies were resolved by a third investigator. The data collected encompassed several aspects, including the first author’s name; study characteristics, including sample size and setting; patient demographics, including age and sex; sensitivity and specificity values; details on the perfusion index; the number of patients exhibiting hypotension; and the country of origin for each study. Attempts were made to contact the authors of the articles to complete any missing information.

### 2.5. Outcomes and Definitions

The primary aim of this meta-analysis was to assess the diagnostic accuracy of the perfusion index in predicting the probability of perioperative hypotension following spinal anesthesia. The criteria for defining hypotension were based on those used in individual studies.

### 2.6. Quality Assessment

Study quality in terms of diagnostic accuracy was assessed using the Quality Assessment for Diagnostic Accuracy Studies-2 (QUADAS-2) tool [19]. This tool encompasses two primary categories: “risk of bias” and “applicability concerns.” The “risk of bias” category consists of four domains, whereas the “applicability concerns” category encompasses three. Both authors independently conducted a subjective review of all included studies to ensure consistency and reliability. Each domain within the QUADAS-2 tool was evaluated and assigned a rating of “low risk”, “some concerns”, or “high risk” by the authors. In cases where discrepancies arose, the authors engaged in thorough discussions to reach consensus. If required, a third author was consulted to resolve any disagreement.

### 2.7. Statistical Analysis

The area under the curve (AUC) was calculated from a summary receiver operating characteristic (ROC) curve, which is widely recognized as a comprehensive measure of the overall accuracy of diagnostic tests. Additionally, the post-test probability was examined using Fagan’s nomogram, a graphical tool that combines the pre-test probability of a disease with the likelihood ratios (LRs) of the test results, enabling estimation of the post-test probability [20]. A positive LR between 2 and 5 slightly increased the likelihood of having the disease after the test. Ratios between 5 and 10 moderately increase the chance, whereas ratios above 10 greatly increase the chance of having the disease [21]. For example, a positive LR (LR+) of 4.0 indicates that a positive test result is four times more likely to occur in a diseased individual than in a healthy individual. Similarly, a negative LR (LR−) compares the chances of acquiring a negative test result when the characteristic being studied is absent to when it is present. To assess potential publication bias, Deek’s funnel plot was employed, which is a graphical approach that investigates the relationship between the standard error and effect size of each study. To mitigate potential bias and heterogeneity across studies, a subgroup analysis was conducted, focusing on studies that utilized similar dosages of local anesthetics and cutoff values of the perfusion index for predicting hypotension. To determine statistical significance, a significance level of 0.05 was adopted. All statistical analyses were performed using the MIDAS command in Stata 15 (StataCorp LLC., College Station, TX, USA).

## 3. Results

### 3.1. Study Selection and Study Characteristics

The study commenced with a systematic and thorough search of four electronic databases, yielding 166 records (Figure 1). After removing duplicate records (*n* = 10) and conducting title and abstract screening, 26 articles were identified as potentially meeting the predefined inclusion criteria. Subsequently, a detailed examination of full-text articles resulted in the exclusion of 14 studies. Ultimately, 12 studies successfully satisfied the established criteria and were included in the review [15,17,18,22,23,24,25,26,27,28,29,30].

The characteristics of the 11 studies involving 2009 patients are summarized in Table 1. This meta-analysis included 12 studies conducted in various countries, including India (*n* = 7) [17,22,23,24,25,27,29], Nepal (*n* = 2) [28,30], Japan (*n* = 1) [15], and China (*n* = 2) [18,26]. All studies focused on parturients who underwent elective cesarean section. The age range of the participants in these studies was 20–35 years old. The number of patients enrolled in each study varied from 35 to 1071 participants. Among the 12 studies, 11 used bupivacaine (range: 9–15 mg) as the anesthetic agent for spinal anesthesia [15,17,22,23,24,25,26,27,28,29,30], whereas one used ropivacaine (12 mg) [18]. The cutoff value for the performance index (PI) varied across studies, ranging from 1.75 to 5.845. However, one study did not provide any specific information. The incidence of hypotension, a commonly observed complication during spinal anesthesia, varied among the included studies, ranging from 13.7 to 76.7% (median: 53.75%).

A comprehensive assessment of the risk of bias and applicability concerns in the 12 included studies is shown in Figure 2. Regarding the risk of bias domains, all studies were rated as low-risk in terms of patient selection, reference standard, flow, and timing. However, three studies raised concerns regarding the index test domain [18,24,26]. Regarding applicability concerns, all studies were categorized as low-risk in the patient selection, index test, and reference standard domains.

### 3.2. Outcomes

To calculate the pooled sensitivity and specificity for predicting hypotension development, 12 studies were included in the analysis [15,17,18,22,23,24,25,26,27,28,29,30]. A summary of the definition of hypotension is presented in Table 1. The pooled sensitivity and specificity values were 0.81 (95% confidence interval CI = 0.72–0.87) and 0.75 (95% CI = 0.67–0.82), respectively, with I^2^ values of 89.43% for sensitivity and 76.35% for specificity (Figure 3) [15,17,18,22,23,24,25,26,27,28,29,30]. The pooled AUC was calculated as 0.84 (95% CI = 0.81–0.87) (Figure 4a). Fagan’s nomogram plot, shown in Figure 5a, displays post-test probabilities based on the LR values. LR+ was 3, indicating a slight increase in the probability of a positive result, whereas LR− was 0.26, indicating a small decrease in the probability of a negative result (Figure 5a). Furthermore, Deek’s funnel plot asymmetry test suggested that the included studies exhibited significant bias in reporting their results (*p* = 0.02) (Figure 6a).

Eight studies that utilized similar dosages of local anesthetics and cutoff values of perfusion index (i.e., 3.5) for predicting hypotension were analyzed [15,17,22,25,27,28,29,30]. The combined sensitivity and specificity values were determined to be 0.74 (95% CI = 0.66–0.81, I^2^ = 53.32) and 0.79 (95% CI = 0.73–0.83, I^2^ = 39%), respectively. The pooled AUC was calculated as 0.83 (95% CI = 0.80–0.86) (Figure 4b). Fagan’s nomogram plot (Figure 5b) showed LR+ and LR− values of 3 and 0.33, respectively. Deek’s funnel plot asymmetry test demonstrated a low risk of bias in reporting results (*p* = 0.77) (Figure 6b).

## 4. Discussion

In an analysis of 12 studies involving 2009 parturients, the pooled sensitivity and specificity for predicting hypotension development were 0.81 and 0.75, respectively. The pooled AUC was 0.84, indicating good predictive accuracy. Fagan’s nomogram plot showed that a positive test result slightly increased the probability of hypotension, whereas a negative test result slightly decreased it. Deek’s funnel plot asymmetry test suggested potential publication bias among the included studies.

Spinal-anesthesia-induced hypotension is a well-known complication associated with the use of spinal anesthesia during surgical procedures. This hypotensive response can be attributed to preganglionic sympathetic fiber blockade, which results in vasodilation, decreased preload, and reductions in cardiac output and mean arterial pressure. Furthermore, baseline vascular tone has been identified as a significant factor in determining susceptibility to hypotension [15]. During pregnancy, a significant decrease in vascular tone due to hormonal, metabolic, and physiological adaptations is observed [31,32,33]. This change in vascular tone facilitates the accommodation of increased blood volume [31,32,33], which is crucial for maintaining an adequate blood supply to the growing fetus and maternal organs [33]. However, a decrease in vascular tone during pregnancy contributes to blood pooling in the extremities, and the induction of sympathectomy by spinal analgesia further exacerbates this pooling, thereby leading to more blood being trapped in the extremities.

Several studies have reported the effectiveness of the perfusion index, an indirect measure of vascular tone, in predicting post-spinal hypotension development [15,17,18]. This meta-analysis was the first comprehensive study to evaluate the effectiveness of the perfusion index in predicting hypotension. By combining evidence from various studies, we provide a robust assessment of the overall efficacy of the perfusion index as a predictive tool. This novel contribution enhances our understanding of the potential utility of the perfusion index in identifying parturients at risk of developing hypotension following spinal anesthesia. The pooled sensitivity and specificity values of 0.81 and 0.75, respectively, suggest that the perfusion index has relatively high accuracy in identifying hypotension occurrence. Furthermore, the pooled AUC of 0.84 indicates a good overall predictive accuracy of the perfusion index in distinguishing between patients who will develop hypotension and those who will not. These findings highlight the potential benefit of incorporating the perfusion index as a predictive tool in this clinical setting.

In this meta-analysis, the incidence of post-spinal hypotension varied from 13.7 to 76.7% (median: 53.75%), likely due to differences in anesthetic dosage and the definition of hypotension. The inclusion of the perfusion index as a diagnostic test demonstrated its potential benefit in improving the identification of parturients under spinal anesthesia. According to the Fagan’s nomogram plot in the current meta-analysis (Figure 5), assuming a pre-test probability of 50%, a positive result from the perfusion index test increased the post-test probability to 76%, whereas a negative result decreased it to 21%. This suggests that the use of the perfusion index could enhance the chances of correctly identifying parturients at risk of post-spinal hypotension by approximately 25%. Notably, while baseline vascular tone is a contributing factor, the occurrence of post-spinal hypotension is primarily determined by sympathetic blockade. Nevertheless, the perfusion index offers a noninvasive and easily measurable parameter for indirectly assessing vascular tone and guiding prophylactic interventions to optimize hemodynamic stability following spinal anesthesia.

A previous study involving 100 patients showed that the analgesia nociception index (ANI), which is calculated based on heart rate variability analysis, is predictive of maternal hypotension following spinal anesthesia [34]. However, the sensitivity and specificity of this method are reportedly 80% and 34%, respectively [34], with an AUC of 0.63. These findings indicate a moderate level of performance with a relatively high rate of false positives and false negatives. Therefore, ANI may not be highly reliable for accurately identifying the occurrence of post-spinal hypotension. In a study involving 35 patients undergoing cesarean section under spinal anesthesia, it was observed that the preanesthetic carotid artery corrected blood flow time (FTc) had a good predictive value for post-spinal hypotension, with an AUC of 0.905 and high sensitivity (90.5%) and specificity (85.7%) [35]. Despite these promising findings, the accurate prediction of hemodynamic status using FTc may pose challenges, particularly for novice sonologists [36,37]. These challenges could potentially impede the clinical applicability and wider adoption of these parameters as reliable predictors of hemodynamic status. In contrast, the perfusion index provides a readily available and noninvasive measure that can be easily obtained during routine monitoring, potentially facilitating its practical implementation in clinical settings.

Analysis of data from 503 women who received spinal anesthesia for cesarean section showed that body mass index (BMI) > 29 kg/m^2^, age ≥ 35 years, and sensory block height (i.e., >Th6 dermatome) were independently associated with hypotension, defined as either a decrease in mean arterial blood pressure of > 20% from baseline or a systolic arterial blood pressure of < 90 mmHg [38]. Moreover, in an analysis of 410 parturients who underwent cesarean section under spinal anesthesia, factors such as newborn weight ≥ 4 kg, baseline systolic blood pressure < 120 mmHg, sensory block height >Th6 dermatome, time interval between spinal induction and skin incision > 6 min, and anesthetist experience were identified as being associated with an increased risk of hypotension [39]. In a study involving 1071 women who underwent cesarean section, factors such as BMI ≥ 30 kg/m^2^, infant weight ≥ 3500 g, and a large dose of local anesthetics were identified as risk factors for intraoperative hypotension [26]. We suggest that incorporation of the perfusion index and other risk factors may improve the accuracy of hypotension prediction. Further studies are necessary to evaluate the feasibility and effectiveness of integrating the perfusion index and these risk factors into predictive models for hypotension with the aim of improving patient outcomes and safety during cesarean sections.

The presence of publication bias and significant heterogeneity among the studies could potentially compromise the reliability and generalizability of our findings. In the current meta-analysis, a significant proportion of the studies (8 out of 12) consistently adopted a perfusion index value of 3.5 to predict hypotension. The incorporation of a study employing a higher dosage of local anesthetics, such as bupivacaine at 15 mg, and utilizing a perfusion index value of 5.845 may potentially introduce a risk of bias [26]. In light of these limitations and to strengthen the validity of our results, we adopted an alternative approach by conducting subgroup-based studies that focused on subsets of studies using similar dosages of local anesthetics and a consistent perfusion index cutoff value of 3.5 for predicting hypotension. The low risk of publication bias and minor heterogeneity in the subgroup studies supports the reliability of our approach.

Based on the subgroup analysis, we suggest an algorithm to efficiently predict and manage post-spinal hypotension using the perfusion index (Figure A1). The algorithm adopts a systematic approach that includes a thorough preoperative assessment, establishment of a target perfusion index value (specifically 3.5), implementation of appropriate interventions in response to low perfusion index values (e.g., fluid challenge), and continuous monitoring during anesthesia to ensure prompt management of hypotension. It is important to note that, while the algorithm provides a standardized framework, its successful implementation relies on adaptation to specific institutional protocols and adherence to evidence-based practices supported by clinical trials.

This meta-analysis has several limitations that should be acknowledged as they can potentially impact the interpretation and generalizability of our findings. First, the variability in the cutoff value of the perfusion index and the different definitions of hypotension among the included studies introduced heterogeneity, which may have limited the comparability of the findings. Second, umbilical artery pH plays a crucial role in assessing neonatal conditions at birth by reflecting both the metabolic and respiratory components of fetal acidemia [40]. It provides valuable information on the acid–base balance and overall fetal well-being during the critical perinatal period. The lack of umbilical artery pH analysis in most of the included studies prevented assessment of the benefit of the perfusion index in predicting hypotension. Third, the correlation between perfusion index and blood pressure throughout the procedure was not measured serially in most studies. The absence of continuous monitoring between these variables hinders our understanding of their dynamic interplay and limits our ability to draw conclusions about their relationships over time.

## 5. Conclusions

Our analysis of 12 studies involving 2009 patients indicated that the perfusion index showed good predictive accuracy for hypotension, with a pooled sensitivity and specificity of 0.81 and 0.75, respectively. These findings suggest that the perfusion index may be a useful tool for predicting the development of post-spinal hypotension. However, it is important to consider heterogeneity among the included studies and potential publication bias. Our subgroup analysis indicated that the perfusion index with a cutoff value of 3.5 hold practical value and relevance for clinical practice. Further research is necessary to validate these findings and explore the practical applications of the algorithm.

## Figures and Tables

**Figure 1 diagnostics-13-02584-f001:**
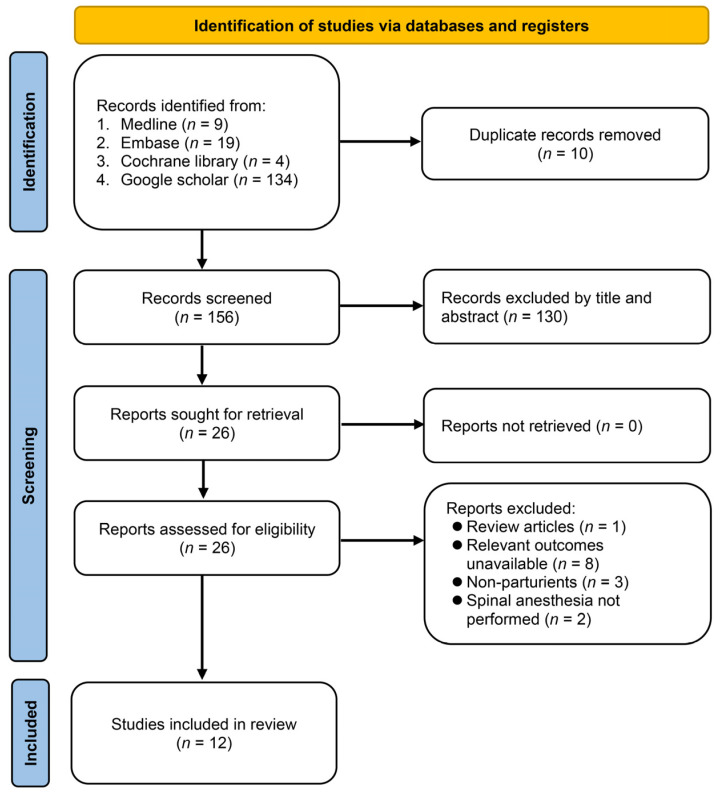
Flow diagram.

**Figure 2 diagnostics-13-02584-f002:**
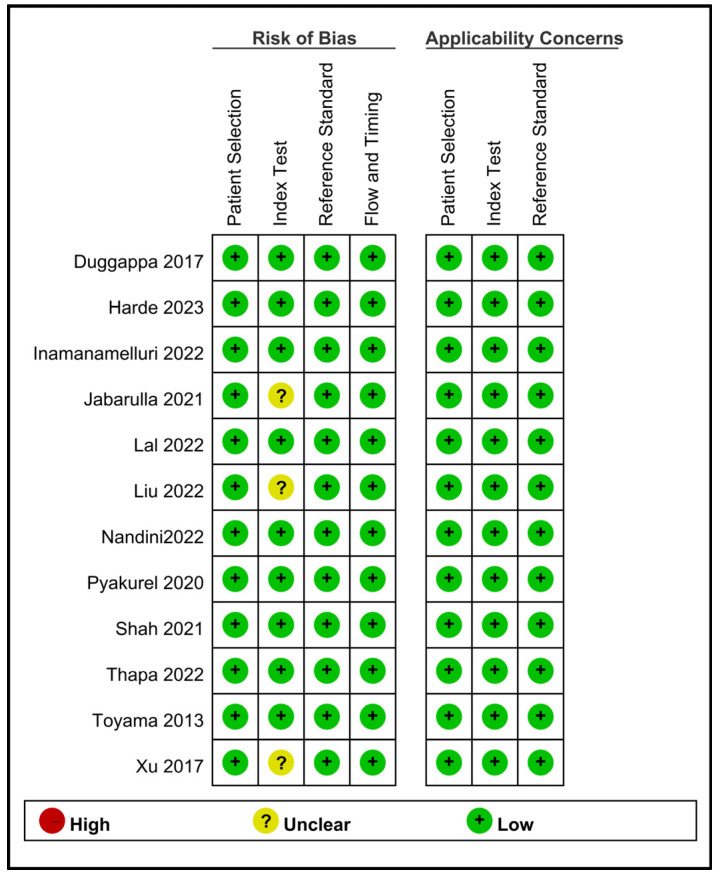
An overview of the methodological quality of the 12 included studies [15,17,18,22,23,24,25,26,27,28,29,30].

**Figure 3 diagnostics-13-02584-f003:**
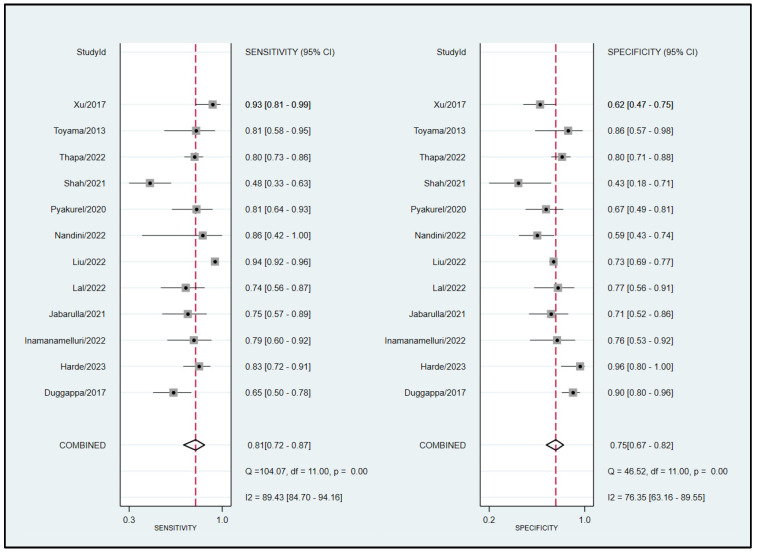
Forest plot illustrating the pooled sensitivity and specificity of the perfusion index (PI) for predicting post-spinal hypotension [15,17,18,22,23,24,25,26,27,28,29,30].

**Figure 4 diagnostics-13-02584-f004:**
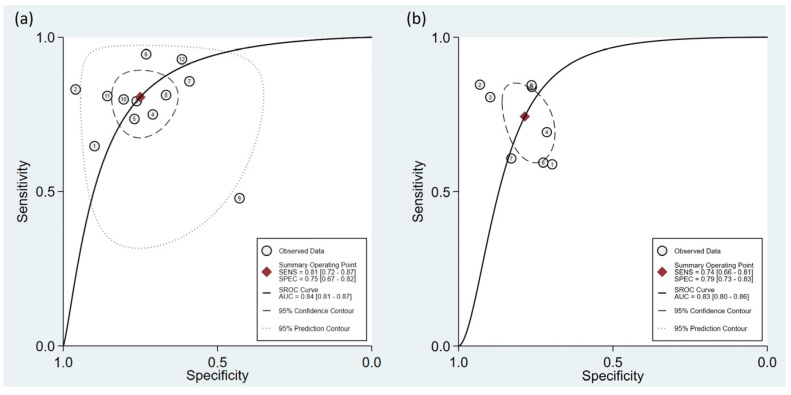
Summary receiver operating characteristic (sROC) curve analysis illustrating the efficacy of perfusion index (PI) in predicting post-spinal hypotension. Analyses were performed based on (**a**) overall studies [15,17,18,22,23,24,25,26,27,28,29,30], and (**b**) studies that utilized similar dosages of local anesthetics and cutoff values of PI [15,17,22,25,27,28,29,30]. The weighted sROC curve is represented by the solid line. Individual study estimates of sensitivity and (1-specificity) are depicted by open circles. Pooled point estimates of outcomes are represented by diamonds, indicating the combined results across studies. AUC refers to the area under the curve, whereas SENS and SPEC correspond to the sensitivity and specificity, respectively.

**Figure 5 diagnostics-13-02584-f005:**
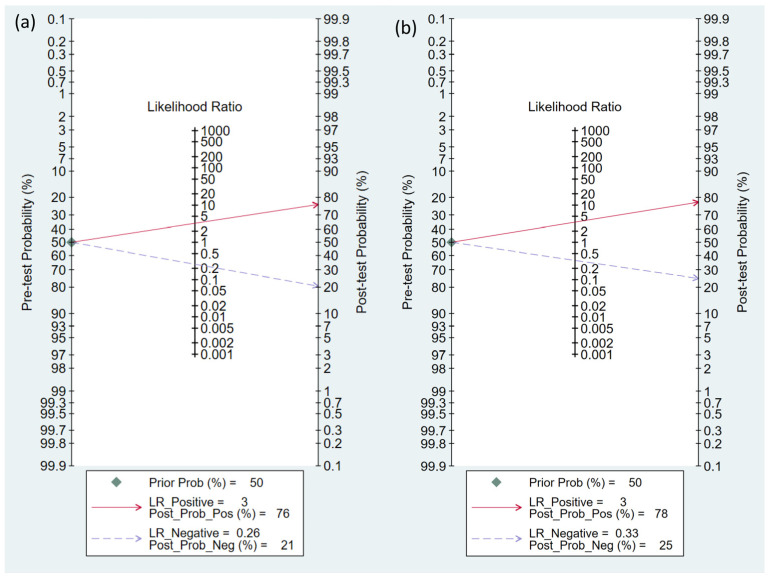
Fagan’s nomogram plot demonstrating the clinical utility of the perfusion index (PI) in hypotension prediction following spinal anesthesia. Analyses were performed based on (**a**) overall studies [15,17,18,22,23,24,25,26,27,28,29,30], and (**b**) studies that utilized similar dosages of local anesthetics and cutoff values of PI [15,17,22,25,27,28,29,30]. LR, likelihood ratio; Prob, probability; Pos, positive; Neg, negative.

**Figure 6 diagnostics-13-02584-f006:**
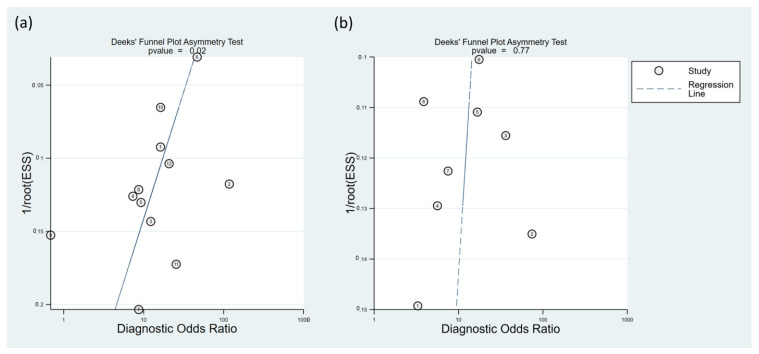
Deek’s funnel plot asymmetry test was performed to evaluate the presence of publication bias across the included studies. Analyses were performed based on (**a**) overall studies (*p* = 0.02) [15,17,18,22,23,24,25,26,27,28,29,30], and (**b**) studies that utilized similar dosages of local anesthetics and cutoff values of PI (*p* = 0.77) [15,17,22,25,27,28,29,30].

**Table 1 diagnostics-13-02584-t001:** Characteristics of studies (*n* = 12).

Studies	Age (Years)	BMI or Weight *	N	Drug (Dosage)	Cutoff PI Values	Definition of Hypotension	IH	AUC	Country
Duggappa 2017 [17]	25 vs. 24	67 vs. 68	120	BUP 10 mg	3.5	MAP < 65 mmHg	42.5%	0.848	India
Harde 2023 [22]	25	58	90	BUP 10 mg	3.5	MAP < 65 mmHg	72.2%	0.917	India
Inamanamelluri 2022 [23]	29 vs. 30	68 vs. 67	50	BUP 10 mg	2.85	MAP < 60 mmHg or <20% of baseline	60%	0.888	India
Jabarulla 2021 [24]	20–35 †	79	63	BUP 9 mg	1.75	<25% of baseline	50.8%	0.78	India
Lal 2022 [25]	26 vs. 25	66 vs. 65	60	BUP 10 mg	3.5	SBP < 80 or <20% of baseline	56.7%	0.74	India
Liu 2022 [26]	29	27.8	1071	BUP 15 mg	5.845	SBP < 90 or <20% of baseline	44.1%	0.8333	China
Nandini2022 [27]	27 vs. 25	64 vs. 62	51	BUP 10 mg	3.5	SBP < 25% of baseline	13.7%	0.799	India
Pyakurel 2020 [28]	26 vs. 24	68 vs. 70	68	BUP 10 mg	3.5	MAP < 20% of baseline	47.1%	0.734	Nepal
Shah 2021 [29]	28 vs. 28	62 vs. 61	60	BUP 10 mg	3.5	MAP < 65 mmHg	76.7%	0.529	India
Thapa 2022 [30]	26 vs. 30	70 vs. 69	247	BUP 11 mg	3.5	MAP < 20% of baseline	60.6%	0.908	Nepal
Toyama 2013 [15]	33	63	35	BUP 10 mg	3.5	SBP < 25% of baseline	60%	0.87	Japan
Xu 2017 [18]	33 vs. 33	28 vs. 27	94	ROP 12 mg	2.2	SBP < 80 mmHg	44.7%	0.81	China

BMI, body mass index; † range, * kg/m^2^ or kg; PI, perfusion index; BUP, bupivacaine; ROP, ropivacaine; AUC: Area under curve; MAP, mean arterial pressure; SBP, systolic blood pressure; IH, incidence of hypotension.

## Data Availability

The original contributions presented in this study are included in the article, and further inquiries can be directed to the corresponding authors.

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
