# Peer review of "Predictive Efficacy of the Perfusion Index for Hypotension following Spinal Anesthesia in Parturient Undergoing Elective Cesarean Section: A Systematic Review and Meta-Analysis"

_diagnostics, 2023, doi:10.3390/diagnostics13152584_

Round 1

Reviewer 1 Report

I congratulate the authors on an extremely well conducted study.  They have done and excellent job of reviewing the papers and using all of the guidelines suggested for insuring repeatable and solid results.  An excellent job all the way around.  There were some very minor issues with grammar (things like a singular pronoun for a group etc.  

Overall very well written.  

Author Response

Response:

We sincerely appreciate the reviewer's comments. To address this concern effectively, we have taken the suggestion seriously and engaged a native speaker to meticulously edit and refine the grammar throughout the paper.

Reviewer 2 Report

Dear authors,

I appreciate your manuscript for quality and essential summary. No significant observation regarding methods, statistical analysis, and results of your tests, but naturally, such a real life challenge as predicting postspinal hypotension in parturients, evoked following questions especially about conclusions.

1- Was it impossible to obtain a cut-off value from available data throughout sROC? Did you evaluate in this way other tests like regression models or cluster analysis?

2- Concerning results of Forest plot, could wide sample size of Liu et al. invalidate your results, due to higher dosages of local anesthetics used (50% more) compared to other studies with smaller samples?

3- If you share the presence of this bias, Deek's funnel plot could reveal a stronger importance in interpretation of results although you suggested a correct way to evaluate Pefusion Index throughout a logic approach to its value.

4- In conclusion, a summary flow chart suggestive for the use of PI in this population could definitely remove all doubts.

Thanks a lot for this involvement.

Author Response

Response to Reviewer2’ comments

General comment:

I appreciate your manuscript for quality and essential summary. No significant observation regarding methods, statistical analysis, and results of your tests, but naturally, such a real life challenge as predicting postspinal hypotension in parturients, evoked following questions especially about conclusions.

Response:

We would like to take this opportunity to express our gratitude to the Reviewer for taking valuable time to review our manuscript. The reviewer’ valuable input and thoughtful review have been instrumental in strengthening our study. We acknowledge that predicting postspinal hypotension in parturients is a real-life challenge, and we appreciate your interest in our research. We understand that the topic raises several important questions, particularly regarding the publication bias and high heterogeneity among studies. In compliance with the professional suggestions of the Reviewer, we have revised our manuscript accordingly. We have also rephrased our conclusions to explicitly highlight the key insights gained from our study and their potential implications for clinical practice. Please find below our point-by-point responses. Please also kindly note that the corresponding changes in the revised manuscript are marked in red.

Comment 1:

Was it impossible to obtain a cut-off value from available data throughout sROC? Did you evaluate in this way other tests like regression models or cluster analysis?

Response 1:

We would like to thank the reviewer for the insightful comments on our study. We agree that obtaining a cut-off value from available data throughout the summary Receiver Operating Characteristic (sROC) curve analysis could provide valuable information to readers. However, in our specific case, we encountered challenges related to publication bias and major heterogeneity among the studies, which could potentially affect the reliability and generalizability of the cut-off value.

Given these limitations and to ensure the robustness of our findings, we decided to pursue an alternative approach. We conducted subgroup-based studies, focusing on subsets of studies that utilized similar dosages of local anesthetics and cut-off values of Perfusion Index (PI), specifically, a cut-off value of 3.5 for predicting hypotension. By grouping studies with similar methodologies, we aimed to minimize the impact of heterogeneity and publication bias on our analysis. It is noteworthy that a considerable number of studies (eight out of 12) adopted the PI value of 3.5 to predict hypotension. This consistency in the use of the same cut-off value across a substantial portion of the studies further supports the validity of our approach. The results of these subgroup-based studies demonstrated a low risk of bias and minor heterogeneity, indicating that the findings are more likely to be informative and reliable for readers. Relevant information have been updated in the method (line 174), result (line 241), and discussion section (line 365-376)) of the revised manuscript.

While we acknowledge that obtaining a single cut-off value from the sROC analysis might be ideal under certain circumstances, the presence of publication bias and major heterogeneity in our dataset warranted an alternative, more robust approach. We believe that the subgroup-based studies have provided valuable insights into the predictive ability of PI with a cut-off value of 3.5 for hypotension and offer practical implications for clinical decision-making. We have thoroughly addressed these concerns, and we hope that our revised approach and analysis will meet the expectations of the reviewer and improve the quality and reliability of our study.

Comment 2:

Concerning results of Forest plot, could wide sample size of Liu et al. invalidate your results, due to higher dosages of local anesthetics used (50% more) compared to other studies with smaller samples?

Response 2:

We would like to thank the Reviewer for raising this concern about the potential impact of the large sample size of the study conducted by Liu et al. on our results. We appreciate the opportunity to address this issue and clarify our approach. We completely understand the importance of considering the impact of variations in sample sizes and local anesthetic dosages across different studies, as these factors can significantly influence the overall results and conclusions of a meta-analysis.

To address this concern, we conducted a subgroup analysis that specifically focused on studies with similar dosages of local anesthetics and Perfusion Index (PI) values. In this subgroup analysis, we excluded the study by Liu et al. and only considered studies that utilized comparable dosages of local anesthetics and had a consistent cut-off value of PI (i.e., 3.5) for predicting hypotension. By excluding the study by Liu et al. from our subgroup analysis, we aimed to minimize any potential influence it might have on our overall findings. This approach allowed us to isolate and analyze the studies that shared similar characteristics, thus reducing the confounding effect of variations in sample size and local anesthetic dosages. Through this subgroup analysis, we were able to evaluate a more homogenous set of studies, which enhanced the internal validity and reliability of our results. Relevant information have been updated in the method (line 174), result (line 241), and discussion section (line 365-376)) of the revised manuscript.

Comment 3:

If you share the presence of this bias, Deek's funnel plot could reveal a stronger importance in interpretation of results although you suggested a correct way to evaluate Pefusion Index throughout a logic approach to its value.

Response 3:

The Reviewer’s insightful comment is highly appreciated. We acknowledge that publication bias can impact the interpretation of results and may affect the overall robustness of our findings. Regarding our subgroup analysis, which was specifically designed to focus on studies utilizing similar dosages of local anesthetics and Perfusion Index (PI) values, we found encouraging results. This approach allowed us to demonstrate a low risk of publication bias within the selected subgroup studies. Importantly, results from subgroup analysis did not significantly influence the diagnostic efficacy of the perfusion index. We thank the reviewer for the thoughtful comments and are committed to continually improving the rigor and interpretability of our research to better serve the scientific community and enhance patient care. Relevant information have been updated in the method (line 174), result (line 241), and discussion section (line 365-376)) of the revised manuscript.

Comment 4:

In conclusion, a summary flow chart suggestive for the use of PI in this population could definitely remove all doubts.

Response 4:

The Reviewer’s important comment is sincerely appreciated. We completely agree that a visual representation can be a powerful tool to convey the key steps and decision points in our approach. In response to Reviewer’s suggestion, we have created a summary flow chart that outlines the algorithm in a clear and concise manner (Appendix 2). The flow chart highlights the sequential steps, including the preoperative assessment, baseline PI measurement, establishment of the target PI value (3.5), interventions for low PI values, and continuous monitoring after spinal anesthesia. By presenting the algorithm in this visual format, we may provide a user-friendly guide that can be readily incorporated into clinical practice. Relevant information have been updated in the discussion section (line 377-386), conclusion section (line 409), and appendix 2 (line 425)of the revised manuscript.
